# Structural and Functional Characteristics of Hemp Protein Isolate–Pullulan Polysaccharide Glycosylation Conjugate in an Aqueous Model System

**DOI:** 10.3390/foods12071416

**Published:** 2023-03-27

**Authors:** Ziwen Ding, Fan Jiang, Kun Liu, Fangshuo Gong, Yuanfa Liu, Zhaojun Zheng, Yong-Jiang Xu

**Affiliations:** State Key Laboratory of Food Science and Technology, School of Food Science and Technology, National Engineering Research Center for Functional Food, National Engineering Research Center of Cereal Fermentation and Food Biomanufacturing, Collaborative Innovation Center of Food Safety and Quality Control in Jiangsu Province, Jiangnan University, 1800 Lihu Road, Wuxi 214122, Chinazhaojun064@163.com (Z.Z.)

**Keywords:** hemp protein isolate, pullulan, glycosylation, techno-functional properties, protein–polysaccharides complex

## Abstract

Hemp protein, with its important nutritional and industrial value, has trickled into the aisles of protein demand; however, its poor functional properties have largely limited its implementation in food. Herein, we aimed to modify hemp protein isolate (HPI) via glycosylation coupling with pullulan polysaccharide, and we subsequently characterized its structural and functional properties. The conjugation variables were HPI to pullulan ratio (i.e., 3:1, 2:1, 1:1, 1:2, and 1:3 *w*/*w*), incubation temperature (i.e., 50, 60, 70, 80, and 90 °C), and incubation time (i.e., 3, 6, 12, 24, and 48 h). Native HPI was used as a control for comparison purposes. We found that DG tended to decrease when the pullulan to HPI ratio was greater than 1:1 and when the temperature exceeded 80 °C. SDS-PAGE analysis shows that when the DG is increased, wider and heavier molecular weight bands emerge near the top of the running gel, while such observations were absent in the control. Further, glycosylation could loosen the HPI’s secondary and tertiary structures, as well as increase surface hydrophobicity. The solubility of HPI after glycosylation significantly increased (*p* < 0.05) at pH 7.0 compared to HPI without glycosylation. Emulsifying activity improved significantly (*p* < 0.05), with glycosylation with HPI–pullulan at a ratio of 1:3 showing maximum emulsifying activity of 118.78 ± 4.48 m^2^/g (HPI alone: 32.38 ± 3.65 m^2^/g). Moreover, the HPI–pullulan glycosylation time of 24 h showed maximum foaming activity (23.04 ± 0.95%) compared to HPI alone (14.20 ± 1.23%). The foaming stability of HPI (79.61 ± 3.33%) increased to 97.78 ± 3.85% when HPI–pullulan was conjugated using a glycosylation temperature of 80 °C. Compared with the un-glycated HPI, HPI–pullulan also increased WHC (4.41 ± 0.73 versus 9.59 ± 0.36 g/g) and OHC (8.48 ± 0.51 versus 13.73 ± 0.59 g/g). Intriguingly, correlation analysis showed that protein functional characteristics were significantly and positively correlated with DG. Overall, our findings support the notion that pullulan conjugation provides further functional attributes to the HPI, thereby broadening its potential implementation in complicated food systems.

## 1. Introduction

At present, the demand for food protein is facing a growing trend because the world population is expected to reach about nine billion by 2050. The increasing cost and limited supply of animal protein is closely related to climate change, freshwater depletion, loss of biodiversity, and human health hazards associated with cardiovascular and other diseases [1,2,3]. At the same time, driven by younger and wealthier consumers, the demand for plant-based foods and protein (plant-based meat and dairy products, etc.) in Asia and other regions is growing. Therefore, the search for sustainable and environmentally feasible alternative sources of plant protein is highly encouraged [4,5]. Hemp (*Cannabis sativa* L.) is widely cultivated in China, accounting for 50% of global production [6]. Its protein is usually a by-product of sesame oil, and most of it is used as animal feed or discarded, demonstrating a lack of sufficient development and utilization [6]. It is important that hemp isolate protein is rich in nutrition and has relatively ideal amino acid composition. The hemp protein contains all the essential amino acids (EAAs) of the human body. The EAA with the highest content in hemp protein is leucine, followed by isoleucine and valine. These branched-chain amino acids are closely related to protein metabolism and the establishment of muscle tissue, while the content in hemp protein is equivalent to that of abalone muscle protein [7]. In addition, the EAA characteristics of hemp protein can also be compared with many high-quality proteins, such as casein, whey protein, and soy protein [4,8,9,10]. However, the compact protein structure of hemp protein results in its poor functional characteristics such as solubility and emulsibility, which largely limit its comprehensive implementation in complicated food systems [8,11]. The structure of protein determines the functional properties of protein. A change in functional properties will affect the sensory quality and texture of food. Good functional properties will help improve the use and application range of food. However, the various functional properties of hemp protein isolate do not directly meet processing needs, so it is usually appropriate to modify it.

One of the effective, controllable, and practical approaches for protein modification is the glycosylation reaction, which occurs naturally and spontaneously through the amino group of protein molecules and the carbonyl group of saccharide molecules without any chemical additions [12,13]. The glycosylation reaction is acknowledged to unfold the protein molecular structure and hinder the aggregation of protein molecules, thereby improving the functional characteristics and application ranges of proteins. In recent years, researchers have found that combining proteins with saccharide molecules through the glycosylation reaction can significantly improve the physicochemical and functional properties of proteins in food systems, with solubility, foaming, and emulsification identified as some of the properties that can be improved [14,15,16]. Moreover, the glycosylation complex has been proven to improve the biological characteristics of antioxidation, cytotoxic activities, and antibacterial activity [17,18]. It is also one of the few ways to realize covalent bonds between food ingredients without any risk to consumer health, as this reaction is completed without adding any toxic compounds. In addition, the glycosylation reaction is highly associated with reaction factors such as humidity, temperature, and time [19,20,21]. Accordingly, exploring the effects of reaction conditions on the corresponding glycosylation reaction is indispensable for achieving the desired functional properties of proteins.

Apart from the reaction conditions, saccharide molecules are also well reasoned as the determinable influential factor for forming protein and polysaccharide conjugates via glycosylation reactions. Recent studies in this area have found that larger molecular weight saccharides tend to produce conjugates with better emulsification and stability properties [22,23]. In general, longer saccharide chains increase the thickness of the interfacial film and make it easier to control the stabilization of the glycosylation reaction in a desirable process, which facilitates the expansion of production in the application system [24,25]. At present, glycosylation products have different applications in various fields, such as providing flavor in food, improving protein function, and acting as an emulsifier, food antioxidant, or carrier of bioactive substances [16,24,26,27,28].

Pullulan, a biodegradable water-soluble polysaccharide, has been widely accepted in food for its high solubility, low viscosity, nontoxicity, lack of odor, and lack of taste [29,30]. To our knowledge, however, there is limited information regarding the formation of hemp protein and polysaccharides. A study by Feng et al. was the only one to have used pectin to enhance the emulsifying properties of hemp seed protein, and they proved that the introduction of polysaccharides significantly inhibited the coalescence of HPI emulsion and ensured the integrity of emulsion under environmental stress [31]. Herein, we adopted the glycosylation reaction to prepare hemp protein and pullulan conjugates under various conditions and subsequently evaluated the structural and functional properties of those conjugates. Therefore, the main purpose of this work is to study the effects of reaction conditions on the physicochemical and functional properties of HPI–pullulan conjugates. To achieve this goal, we produced glycosylated HPIs with different substrate ratios, temperatures, and times. The effects of culture conditions on the grafting degree, browning strength, protein structure, and functional properties of HPI–pullulan conjugates were analyzed. We evaluated the relationship between binding levels and different functional properties, providing new ideas for creating effective conditions to modify glycosylation of cannabis protein and expand its use.

## 2. Materials and Methods

### 2.1. Materials

Hemp seeds were purchased from Guangxi, China. Pullulan were purchased from Beijing Innokai Technology Co., Ltd., (Beijing, China). All other reagents used in this study were provided from Sinopharm Chemical Reagent Co., Ltd., (Shanghai, China) and were of analytical grade. while the water used was distilled water.

### 2.2. Preparation of the HPI

HPI was prepared and extracted using the alkali-soluble acid precipitation method according to the method of a previous study [9]. The hemp seeds were ground into powder, and n-hexane was added at a ratio of 1:5 material to liquid (*w*/*v*). The mixture was evenly stirred at room temperature for 6 h to remove the fat and the organic reagent from the supernatant while leaving the degreased hemp powder precipitated in the lower layer. The degreased hemp powder was then mixed with deionized water at a ratio of 1:5 (*w*/*v*) and the pH was adjusted to 10.0 with NaOH, with the mixture then stirred at room temperature for 2 h before being centrifuged at 8000 r/min for 30 min (Sorvall Lynx 4000 centrifuge, Thermo Fisher Scientific Co., Ltd., Waltham, MA, USA) so the supernatant could be collected. The supernatant was adjusted to a pH of 4.8 with HCl. After standing at room temperature for 30 min, the supernatant was centrifuged at high speed (4 °C, 8000 r/min) for 30 min and precipitated. An appropriate amount of water to dissolve the precipitate was added and the pH was adjusted to 7.0 with NaOH and HCl, with the mixture then placed in a refrigerator at −80 °C for pre-freezing. The mixture was then frozen and dried in a lyophilizer (SCIENTZ-20F/A freeze dryer, Ningbo Science and Biotechnology Co., Ltd., Ningbo, China) to form hemp protein isolate (HPI). The protein content of HPI was 92.5 g/100 g as determined by a Kjeldahl analyzer (K9860 automatic Kjeldahl nitrogen analyzer, Haineng Instruments Ltd., Jinan, China), with a conversion factor of 6.25.

### 2.3. Preparation of the HPI–Pullulan Conjugates

HPI was designated as the untreated sample, and all glycosylated samples are collectively referred to as HPI–pullulan. The glycated HPI with pullulan polysaccharide was prepared according to the three following procedures:(1)HPI (3% (*w*/*w*)) and 1, 2, 3, 6, or 9% (*w*/*v*) pullulan was dissolved in phosphate buffer solution (0.2 M, pH 7) and then stirred for 2 h at 20 °C to allow for complete hydration. The specimens were heated at 70 °C for 12 h in an oscillator (desktop thermostatic oscillator, Shanghai Boxun Industrial Co., Ltd., Shanghai, China). The sets of samples were labeled 3:1, 2:1, 1:1, 1:2, and 1:3.(2)HPI (3% (*w*/*w*)) and 3% (*w*/*v*) pullulan was dissolved in phosphate buffer solution (0.2 M, pH 7) and then stirred for 2 h at 20 °C to allow for complete hydration. The specimens were heated at 50 °C to ~90 °C for 12 h in an oscillator. The sets of samples were labeled 50 °C, 60 °C, 70 °C, 80 °C, and 90 °C.(3)HPI (3% (*w*/*w*)) and 3% (*w*/*v*) pullulan was dissolved in phosphate buffer solution (0.2 M, pH 7) and then stirred for 2 h at 20 °C to allow for complete hydration. The specimens were heated at 70 °C for 3~48 h in an oscillator. The sets of samples were labeled 3 h, 6 h, 12 h, 24 h, and 48 h.

After the above glycosylation reaction, the samples were placed in an ice water bath and cooled to stop the reaction immediately. The glycosylated treated HPI samples were obtained by lyophilization and placed in a −80 °C refrigerator.

### 2.4. Determination of Free Amino Groups and Measurement of UV absorbance

Free amine groups were measured using the o-phthaldialdehyde (OPA) assay according to the method of a previous study [32].

The absorbance of glycosylated samples (5 mg/mL) was measured at 420 nm using a spectrophotometer (UV-2401 Shimadzu Corporation, Kyoto, Japan) to verify the level of the final stage of the glycosylation reaction [33].

### 2.5. SDS-PAGE

SDS-PAGE experiments were performed using a discontinuous buffer system with 12% separation and 8% stacking gel. The sample loading volume was 5 μL/lane and gels were stained with Komas Brilliant Blue (R250) for 1–2 h and then decolorized with the aqueous solution for 24 h. The gel was imprinted by a fluorescence imaging system (ChemiDoc MP System, BIO-RAD, Shanghai, China).

### 2.6. Spectroscopy Properties

#### 2.6.1. Circular Dichroism (CD) Spectra

Circular dichroism (CD) spectra were determined by a spectropolarimeter (Chirascan V100, Applied Photophysics, Ltd., Surrey, Britain) according to a previous study [34].

#### 2.6.2. Surface Hydrophobicity

The samples’ surface hydrophobicity was calculated using an F-7000 fluorescence spectrophotometer ( Hitachi, Tokyo, Japan). The instructions for the procedure are as follows: Disperse the sample into phosphate buffer (50 mM, pH 7.4) and dilute the concentration of protein sample to 0.005–2.00 mg/mL. Add 20 µL of 8 mmol/L ANS to 4 mL of protein sample buffer and use a spectrophotometer to determine H_0_. Set the emission wavelength to 470 nm and the excitation wavelength to 390 nm and scan the sample within the wavelength range of 400~600 nm at the emission interval of 5 nm. Calculate the slope of the function of fluorescence intensity versus HPI concentration, namely the surface hydrophobicity index of protein.

#### 2.6.3. Intrinsic Fluorescence Emission Spectroscopy

According to the description of a previous study [35], the sample was scanned using a fluorescence spectrophotometer (F-7000, Hitachi Corporation, Tokyo, Japan). Instructions for the procedure are as follows: Use 10 mM phosphate buffer (pH 7.0) to adjust the protein hydrolysate to the concentration of 0.1 mg/mL. Set the excitation wavelength to 290 nm to excite the sample solution and scan the emission spectrum of the sample in the range of 300 nm–400 nm.

#### 2.6.4. Fourier Transform Infrared Spectroscopy (FT-IR) Analysis

Lyophilized conjugates were ground with KBr at a ratio of 50:1 and then compressed to form discs. The FT-IR spectrum of the samples was recorded on an infrared spectrometer (Thermo Scientific Nicolet iS10 spectrometer, Waltham, MA, USA) over a wavelength range of 500–4000 cm^−1^ with a resolution of 4 cm^−1^ for 32 times accumulation at room temperature.

### 2.7. Functional Properties

#### 2.7.1. Determination of Foaming Properties

Foaming ability (FA) and foam stability (FS) were determined according to a previous method [36]. A total of 25 mL of 10 mg/mL protein solution was poured into a 50 mL measuring cylinder, which was then sheared for 2 min with an Ultra-Turrax18 high-speed disperser at 10,000 rpm/min. The height of foam was measured at 0 min and 30 min.

#### 2.7.2. Protein Solubility (PS) Measurements

Protein solubility was determined by dispersing the samples in distilled water to obtain a final solution of 0.2% (*w*/*w*) in protein. The solution was centrifuged at 2400× *g* for 30 min in a Kubota Model 6700 centrifuge. The amount of soluble protein before and after centrifugation was determined using the Lowry method [37].

#### 2.7.3. Water Holding Capacity (WHC) and Oil Holding Capacity (OHC)

The water and oil holding capacity of HPI was measured according to a previous method [38]. WHC and OHC were calculated from the difference between the final and initial weights of tube content before and after centrifugation.

#### 2.7.4. Emulsifying Properties

Emulsifying ability (EAI) and emulsifying stability (ESI) were measured according to a previous method [39]. EAI and ESI were calculated using the following equations:(1)$$\mathrm{EAI}\left(\frac{m^{2}}{g}\right)=2\times \left(2.303\times \;A500\right)\times \;N\;\times 10-4/\psi \mathrm{LC}\;$$
(2)$$\mathrm{ESI}\left(\min\right)=\frac{A_{0}}{A_{0}-A_{t}}\times 10$$
where EAI is the emulsifying area per gram of protein (m^2^/g), ESI is the emulsion stability index, N is the dilution factor (100), C is the protein concentration (g/mL), L is the optical path (1 cm), and ψ is the oil volume fraction (0.25). A_0_ and A_t_ are the absorbances of the emulsion at 0 min and 30 min, respectively.

### 2.8. Statistical Analysis

All experiments were conducted in triplicate, and the results were expressed as the average standard error (SEM) of the average value. SPSS software (SPSS Inc., Chicago, IL, USA) was used for one-way ANOVA of all data, and the significance level was *p* < 0.05.

## 3. Results and Discussion

### 3.1. Degree of Glycosylation Reaction between Hemp Protein Isolate and Pullulan

Glycation occurs through condensation reaction of the carbonyl group in reducing sugars with free amino groups in proteins to form the Schiff base, and the grafting rate (DG) is often used to characterize the extent of glycosylation reactions. As shown in Figure 1, all HPI–pullulan conjugates presented DG values ranging from 6.85% to 29.33%, indicating the covalency glycation of sugar chain and protein molecules with the progress of the glycosylation reaction. Specifically, a continuous increase was observed in the DG of HPI–pullulan conjugates as a function of pullulan concentrations, where the relatively higher value (27.83%) was achieved at the mass ratio of 1:1 between hemp protein and pullulan (Figure 1A). This phenomenon might be explained by the increasing reaction sites provided by high concentrations of pullulan, thus elevating intermolecular collision binding [40]. However, at the mass ratio of pullulan to HPI of 2:1 or even 3:1, DG content reduction and browning degree were significantly deepened (*p* < 0.05). This may be closely related to the mass fraction of solids in the solution. When the concentration increases to a certain range, the viscosity of the system increases and the steric hindrance of macromolecules reduces the impact probability between molecules, thus affecting the progress of the glycosylation reaction. Thus, we chose the 1:1 ratio for HPI and pullulan for further glycosylation condition experiments. Reaction temperature was found to cause an increase in DG at 70 °C, after which DG declined with the further increase in temperature (Figure 1A). We then analyzed the influence of glycosylation temperature on the 1:1 mass ratio HPI to pullulan sample. However, as the temperature was more than 80 °C, DG decreased. This is because protein macromolecules aggregate at high temperatures. Similar behavior was also observed by Li, Hettiarachchy [41], demonstrating that high temperature leads to the disassociation of glycans from protein molecules. Furthermore, we explored the influence of glycosylation time between HPI and pullulan on their conjugation. We found that a rapid rise in DG was observed in HPI–pullulan conjugates when glycosylation reaction time was increased from 6 h to 12 h (*p* < 0.05). The higher conversion was attributed to acceleration of the chemical reactions involved and greater unfolding of the protein, which resulted in the exposure of more amine-containing residues [16]. However, further increasing glycosylation time did not lead to significant continued increases in DG, which may be due to protein aggregation caused by the long reaction time.

The degree of browning of glycosylated products was adopted to evaluate the endpoint level of the glycosylation reaction using absorbance measurements at 420 nm. As shown in Figure 1B, A420 also continuously increased with increases in the proportion of pullulan polysaccharide in the reaction system (*p* < 0.05). A similar phenomenon was also observed for increasing time and polysaccharide concentration. Therefore, it is also very important to select suitable glycosylation reaction conditions to prevent browning from deepening. In order to further explore the effect of glycosylation reaction conditions on HPI–pullulan, we chose to further explore its structure and functional characteristics.

### 3.2. SDS-PAGE Profiles

The formation of the hemp protein isolate and pullulan polysaccharide conjugate can be determined by SDS-PAGE. The covalent coupling between hemp protein and pullulan was identified and is depicted in Figure 2. A typical electrophoretic profile of HPI shows that the main component of HPI is edestin, which consists of six identical subunits. Each subunit is formed by binding an acidic subunit A and a basic subunit B, respectively. Under reducing conditions, the disulfide bond connecting the subunits breaks, releasing two different bands of 34 kDa and 21 kDa corresponding to the edestin A and B subunits [42]. At the ratio of 3:1 between hemp protein and pullulan, a new band was observed above the 34 kDa protein band, indicating that the protein sequence of HPI had been greatly modified and covalently crosslinked with pullulan polysaccharide to form a higher molecular weight conjugate. Interestingly, after 3 h glycation, no high molecular weight components were observed (Figure 2). This shows that the production of polymer conjugates requires a longer glycosylation time. Moreover, with the increase in polysaccharide proportion, the color of electrophoresis gel at a high DG became lighter, indicating that the reduction in free amino groups affected the binding of Coomassie brilliant blue molecules to protein molecules, thus indirectly proving that the HPI and pullulan polysaccharides had covalently bonded. A similar phenomenon was also observed for increases in glycosylation temperature and time, which is consistent with the findings of Chen, Chen [43].

In addition, a high molecular weight conjugate band also appeared near the boundary between stacking and separating gel (Figure 2, channels 3–17), which can be attributed to the formed protein–polysaccharide copolymer after the glycation of hemp protein and pullulan polysaccharide. The copolymer struggles to enter the separation gel through the concentrated gel due to its large molecular weight, and the high molecular weight of the graft production reduces the electrophoretic rate, leading to accumulation at the concentrated gel and the separation gel. Furthermore, the SDS-PAGE results are consistent with previous findings concerning peanut protein, soybean protein, and ovalbumin [43,44,45,46]. Nevertheless, SDS-PAGE analysis could not reflect the specific conformational differences between the glycosylated samples. We further characterized the glycosylation degree of HPI–pullulan using spectroscopy.

### 3.3. Spectral Characteristics of HPI–Pullulan Conjugate

#### 3.3.1. Fourier Transform Infrared Spectroscopy (FT-IR) Analysis

To deeply understand the functional groups involved in the combination with pullulan polysaccharide, the infrared spectra of samples at different times, temperatures, and proportions were analyzed and are displayed in Figure 3A.

During the glycosylation process, by increasing the mass ratio of pullulan to HPI from 1:3 to 1:1, the N-H stretching vibration peak from 3282.11 cm^−1^ to 3269.99 cm^−1^ (HPI’s N-H stretching vibration peak in 3285.24 cm^−1^) can be observed, demonstrating the success of glycosylation as characteristic structures of the sugar chain are found at vibration peaks of 3300 cm^−1^. The 1:1 mass ratio pullulan to HPI glycosylated sample showed stronger absorption at 3300 cm^−1^ in comparison to the same group of glycosylated samples, indicating that the reducing end of the polysaccharide chain is covalently combined with amino acids, thus causing the stretching vibration of N-H [25].

The former results have proven that pullulan polysaccharide had successfully reacted with HPI to produce glycosylated products. The absorption peak of HPI at 1632.76 cm^−1^ is the characteristic peak of the amide I band, which is caused by C=O stretching vibration and N-H (peptide chain) bending vibration [47]. The peak strongly migrates to 1651.35 cm^−1^ in glycosylated samples at the ratio of 1:1 that underwent reaction at 90 °C for 12 h. Other samples also migrated significantly, demonstrating that the glycosylation reaction had changed the structure of the HPI. Additionally, this migration reflects the formation of the Schiff base containing the C=N structure after glycosylation [47].

In HPI, the amide II band has characteristic absorption at 1515.97 cm^−1^, which is caused by C-N stretching and N-H bending. After the glycosylation reaction, the absorption peak of the sample with a pullulan to protein ratio of 1:1 that underwent reaction at 70 °C for 24 h shifts to 1537.92 cm^−1^, and other samples also shift significantly to varying degrees, representing the increase in amide groups in the protein molecular structure during the glycosylation reaction process. Earlier studies indicated that the conversion of primary amino groups during the glycosylation reaction led to the formation of the amide II band, and the formation of the tertiary amine led to a reduction in peak strength [48]. This may be the reason why the amide II band absorption peak for the HPI–pullulan conjugate is weakened.

Simultaneously, a new absorption peak appeared at 1000–1200 cm^−1^, which was caused by the stretching vibration of C-C, C-O, and C-H in pullulan polysaccharide molecules. The surface pullulan molecules were grafted to the protein molecular structure.

#### 3.3.2. Circular Dichroism Spectra Measurements

In circular dichroism spectra, the position and intensity of absorption peaks can reflect the secondary structure of proteins or peptides. The circular dichroism spectra of HPI and HPI–pullulan conjugates are shown in Figure 3B.

The positive peak at 195 nm and the negative peak at 210 nm of HPI represents the β -fold and α-helix, respectively. The shape of the negative peak is closely related to the structure of the protein or peptide chain. Compared with HPI, the negative peak position of HPI–pullulan conjugates shifted from 210 nm to 200 nm, and peak width became narrower. This indicates that the glycosylation reaction between HPI and pullulan can significantly change protein structure, with HPI mainly presenting a random curled state. Interestingly, DG was 27.83% and exhibited a stronger absorption peak at 200 nm than when DG was 22.45% (Figure 3B), which shows that higher DG will cause further changes in protein structure. Moreover, the same phenomenon occurs in Figure 3B, where glycosylation time increases from 12 h to 48 h (DG decreased from 27.83% to 23.13%). These results may be ascribed to the change in the secondary structure of the protein in glycosylation and may depend on many factors, such as glycosylation time, proportion, and temperature.

#### 3.3.3. Surface Hydrophobicity (H_0_)

Protein surface hydrophobicity is closely related to the functional properties of the protein and is the index of the number of hydrophobic groups on the protein surface in contact with the polar aqueous environment. The effect of the glycosylation reaction with pullulan polysaccharide on the surface hydrophobicity of HPI is shown in Figure 3C. The H_0_ of HPI was only 1870.4, and the H_0_ of all HPI–pullulan conjugates increased markedly (*p* < 0.05). However, surface hydrophobicity decreased with increased polysaccharide to protein ratio (mass ratios of HPI and pullulan from 3:1 to 1:3). It can be concluded that the pullulan polysaccharide shielding effect and the formation of advanced glycosylation products causes surface hydrophobicity to decrease. This is also related to the decreased DG with the increased pullulan proportion in the system, as shown in Figure 1A. It further indicates that the reaction ratio of protein and sugar should be properly selected during glycosylation reactions with polysaccharides.

On the contrary, surface hydrophobicity increased with increasing glycosylation time and temperature. After glycosylation for 48 h, the H_0_ of HPI–pullulan conjugates significantly increased to 5461.47, a 2.92-fold increase compared to the H_0_ of HPI (*p* < 0.05). There was no doubt that glycosylation treatment would expose the inner hydrophobic groups originally buried in the hemp isolate protein molecule [40]. Different glycosylation conditions have different effects on H_0_, such as P. ostreatus β-glucan, which can decrease the surface hydrophobicity of oat protein isolates in conjugate form [49]. This indicates that different glycosylation conditions have different impacts on protein structure.

#### 3.3.4. Intrinsic Fluorescence Analysis

Intrinsic fluorescence can effectively reflect conformational changes in in the tertiary structure of HPI after glycosylation. After excitation at 290 nm, the fluorescence spectrum mainly reflects changes in the polarity of tyrosine and tryptophan, which can more sensitively reflect the changes in protein conformation at the tertiary structure level.

Compared with HPI, the maximum emission wavelength of HPI–pullulan conjugate was shifted from 320 nm to 330 nm (Figure 3D). After increasing the mass ratio of pullulan to HPI, the fluorescence intensity of the protein increased significantly, indicating changes in the tertiary structure of the protein (Figure 3D). This is due to the increase in pullulan polysaccharide chains, which increases the grafting ratio and fluorescence intensity. Figure 3D(b,c) show that the fluorescence intensity of the adducts first increases and then gradually decreases with increasing incubation time and temperature, and the changes in fluorescence intensity may be attributed to changes in DG, the shielding effects of the polysaccharide chains attached to the protein, or interference with fluorescence. The glycosylation reaction exposes more of the tryptophan luminescent groups (such as those located inside the globular structure of proteins) to the solvent and makes them more likely to fluoresce [40]. In addition, displacement of the maximum emission peak of HPI can indicate that the polarity of the fluorescent group in the fire hemp protein molecule has changed and that the tertiary structure of the fire hemp protein has been partially unfolded after glycosylation with pullulan polysaccharide, which changed the original protein structure. This is consistent with the results for the infrared spectrum and CD spectrum changes.

### 3.4. Functional Properties of HPI–Pullulan Conjugate

#### 3.4.1. Solubility

Solubility is the determining factor that allows protein to exert interfacial functions such as emulsification, water retention, and foaming. To define the functional characteristics of HPI–pullulan conjugate, our study analyzed protein solubility (PS) at a neutral pH of 7.0. As shown in Figure 4, the solubility of HPI without glycosylation was 28.43%. After glycosylation modification, the PS of all glycosylated samples increased significantly. When DG increased from 22.77% to 27.83% (HPI and pullulan ratio changed from 3:1 to 1:1), the PS of HPI–pullulan increased from 38.55% to 40.76% (*p* < 0.05), as shown in Figure 4A. This is due to the higher anionic charge and hydrophilicity after conjugation with polysaccharides [50]. In Figure 4B,C, with the increase in glycosylation time and temperature, solubility was proportional to the changing trend in DG. The solubility of proteins can be understood as the balance of protein solvent (hydrophilic) and protein–protein (hydrophobic) interactions. This phenomenon can be attributed to an increase in the number of hydrophilic groups compared to individual proteins [49]. From this perspective, solubility decreases with decreases in DG, which may be because of the excessive exposure of hydrophobic groups caused by long glycosylation reaction times between HPI and pullulan, thus increasing protein–protein interaction, promoting aggregation and precipitation, and correspondingly reducing solubility. In addition, with the extended culture time, the number of sugar chains on the protein surface tends to be stable. Hence, the attached sugar chains are not sufficient to balance the increasing hydrophobic aggregation, leading to further extended glycosylation time and no further improvement in solubility. Aminlari, Ramezani [51] also discovered that better solubility was obtained when samples were incubated for 48 h than for 72 h. Therefore, selecting the appropriate glycosylation time to modify the protein can obtain the best solubility.

#### 3.4.2. Emulsifying Properties

As one of the most important functional properties of proteins, the emulsifying properties of a protein directly affect the texture and sensory quality of food [16]. Emulsification is closely related to the structural changes in hemp protein itself and has a certain relationship with other functional properties. EAI refers to the ability of protein to produce emulsion; ESI refers to the ability of protein to stabilize lotion for a certain time. In this study, the emulsifying ability (EAI) and emulsifying stability (ESI) of HPI and HPI–pullulan conjugates were measured, as shown in Figure 5.

With the change in polysaccharide concentration, an increasing and then decreasing trend was observed, and the emulsification of modified protein was significantly different from that of hemp protein (*p* < 0.05). When the concentration of pullulan was twice that of hemp protein, the EAI of the copolymer was increased from 32.38 m^2^/g to 115.67 m^2^/g and the ESI was increased from 39.79 min to 86.56 min compared to the original HPI. On account of the glycosylation reaction, the molecular weight and steric hindrance of protein molecules were increased due to access to sugar chains. In addition, glycosylated hemp protein has more hydrophilic groups than non-glycosylated hemp protein, which increases the steric hindrance effect. Hydrophobic residues in protein molecules can attach to the surface of oil drops, while sugar molecules can attract water molecules next to oil drops. Therefore, glycosylation is conducive to the ability of the protein to stabilize the oil–water interface and reduce the trend of oil drop aggregation [20]. Accordingly, copolymer complexes exhibit better emulsifying properties and emulsifying stability than HPI [19]. In addition, the increase in solubility accelerates the diffusion rate of protein at the oil–water interface, which further contributes to improving the EAI of HPI. Zhong, Ma [52] reported that the emulsifying capacity of oat protein isolate-conjugated *Pleurotus ostreatus* beta-glucan was significantly improved after glycosylation treatment. Li, Hettiarachchy [41] find that xanthan gum when used to saccharify SPI can also significantly improve EAI. Lactoglobulin and peanut protein isolate also exhibited similar improvements in protein emulsification characteristics after glycosylation [21].

When glycosylation time increased from 12 h to 48 h, ESI decreased from 104.64 m^2^/g to 93.86 m^2^/g. This may be due to the partial aggregation of hemp proteins under the strong hydrophobic interaction when hydrophobic groups are overexposed (Figure 3C), resulting in a decline in emulsifying properties and emulsification stability and a slowly increasing DG. The same phenomenon occurs when glycosylation temperature increases from 70 °C to 80 °C. However, a ratio of 1:2 between hemp protein and pullulan led to higher emulsifying ability than a ratio of 1:1 with a higher DG, and it may be that the ungrafted pullulan polysaccharide plays a certain emulsifying role. What is obvious is that the changing trend in emulsification characteristics is similar to that of DG, which may potentially indicate that DG is closely related to the function of proteins. According to the former results, glycosylation can promote the rapid absorption of hemp protein at the oil–water interface, improving the interfacial characteristics of hemp protein.

#### 3.4.3. Foaming Properties

Foaming property (FC) and foaming stability (FS) are vital interfacial properties of protein molecules. Protein foaming has a significant impact on food processing for foods such as whipped cream and other lotion systems.

Figure 6 shows that the FC and FS of glycosylated protein were significantly different from that of hemp protein (*p* < 0.05), and while FC was originally 14.20% (HPI), it increased to a maximum of 23%. With the increasing mass ratio of pullulan to HPI, glycation degree increased from 6.85% to 27.83% (Figure 1A). Meanwhile, foaming ability increased from 15.79% to 21.98% (*p* < 0.05), as shown in Figure 6A. This phenomenon could be attributed to glycosylation modification introducing the polysaccharide chain and adding a large number of hydroxyl groups to the molecule to increase the flexibility of the protein molecule so that the foam is not easily broken due to the change in protein concept. This trend is similar to emulsibility, and it is further confirmed that a higher DG will lead to better functional features for proteins. In addition, no significant difference was observed for FC between glycosylated samples (*p* > 0.05); however, FC was significantly higher for glycosylated samples than HPI alone (*p* < 0.05). Owing to the untreated hemp protein showing an irregular and compact conformation, it was difficult to wrap gas in the shearing process. The positive effect of glycosylation treatment on enhancing the foaming properties of protein has also been reported in other research. Li et al. [41] observed that SPI and D-glucose conjugate showed better foaming properties than SPI. Hence, after the reaction with pullulan polysaccharide, foaming property and foaming stability achieve significant improvements compared to un-glycated hemp protein. This may expand the utilization of HPI–pullulan in food systems.

#### 3.4.4. Water and Oil Holding Capacity

Water and oil holding capacity (WHC/OHC) reflects the ability of the protein to bind water and oil, which is related to the viscosity, taste, and texture of food [38]. Figure 7 shows the influences of glycosylation treatment on the WHC and OHC of HPI and HPI–pullulan. The WHC of un-glycated HPI is 4.14 g/g. The WHC of HPI–pullulan conjugate gradually increased to a maximum value of 10.18 g/g with the increase in DG. The variety trend of WHC agrees with DG; this phenomenon describes the dissociation and partial dissolution of protein molecules in the process of glycosylation promotes the hydration of proteins.

In addition, OHC increased from 8.48 g/g for the original HPI to 15.69 g/g after glycosylation. Interestingly, OHC and H_0_ showed similar trends, indicating that proteins with higher H_0_ tend to have higher fat absorption ability (Figure 3C). Therefore, the strengthening of OHC due to glycosylation introduces the hydrophilic polysaccharide side chain, allowing the protein molecules to better combine with lipids after unfolding. Similar results were observed in other glycosylated reactive proteins [26].

### 3.5. Correlation Analysis between the Grafting Degree of HPI–Pullulan Conjugate and Its Protein Functional Properties

To further explore the relationship between each functional property of HPI−Pullulan and the degree of glycosylation reaction after glycosylation with pullulan polysaccharide, Pearson correlation analysis was performed and heat maps were drawn for DG and its functional properties. As shown in Figure 8A, under different glycosylation ratios, the DG was positively correlated with EAI, ESI, WHC, OHC, FC, and solubility. The correlation coefficients of DG with OHC, WHC, and solubility were 0.956, 0.914, and 0.973, showing a significant positive correlation (*p* < 0.05). In addition, there was also a correlation between the functional properties of the proteins and the correlation coefficients of solubility and water and oil holding capacity, which were 0.903 and 0.895, respectively; this relationship also exhibited a significant positive correlation (*p* < 0.05). As shown in Figure 8B,C, DG was also positively correlated with EAI, ESI, H_0_, WHC, OHC, FC, and solubility at different glycosylation reaction temperatures and times. This indicates that there is a correlation between the degree of glycosylation and improvements in the functional properties of proteins. To a certain extent, the different functional properties of proteins also have a mutual promotion relationship. Therefore, glycosylation modification of proteins with different degrees of glycosylation and suitable glycosylation conditions can be selected to meet the needs of practical applications and production.

## 4. Conclusions

In this study, the protein structure characteristics and physiochemical properties of HPI and HPI–pullulan underwent comprehensive investigation. The results show significant differences in the structures and physiochemical properties obtained between HPI and HPI–pullulan. Despite all HPI−Pullulan samples being successfully conjugated by different glycosylation reactions, they presented different reactivity. SDS-PAGE and spectroscopy results showed that HPI–pullulan changed the structure of secondary and tertiary proteins after forming a higher molecular weight conjugate. Pearson correlation analysis showed that DG was positively correlated with the functional characteristics of hemp protein. The results show that the optimal treatment conditions for HPI−Pullulan (at a 1:1 ratio of HPI:Pullulan) are 12 h at 70 °C, with these conditions allowing for HPI−Pullulan conjugates with excellent functional properties to be efficiently obtained. In conclusion, these findings not only help to guide the development of hemp protein as an ingredient in complex foods such as meat products, dairy products, and desserts, but also have important significance in terms of how to control the glycosylation of polysaccharides to achieve an accurate design of enhanced hemp protein with excellent functional properties.

## Figures and Tables

**Figure 1 foods-12-01416-f001:**
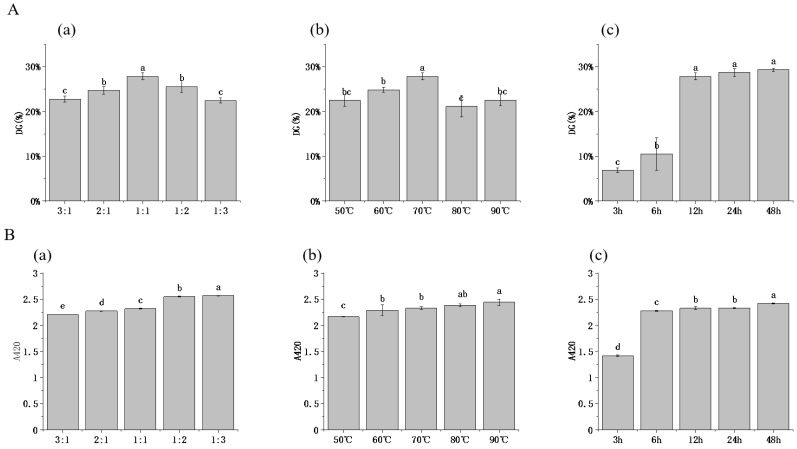
Effect of pullulan, ratios to hemp protein isolate (HPI), incubation temperature, and incubation time on the DG (**A**)and A420 values (**B**) of glycosylated HPI. (**a**): material feeding ratio (pH 7.0 and heated at 70 °C for 12 h); (**b**): reaction tempetature (pH 7.0 and heated 12 h with a HPI to pullulan ratio of 1:1); (**c**) reaction time (pH 7.0 and heated at 70 °C with a HPI to pullulan ratio of 1:1). Different superscript letters represent a significant difference (*p* < 0.05).

**Figure 2 foods-12-01416-f002:**
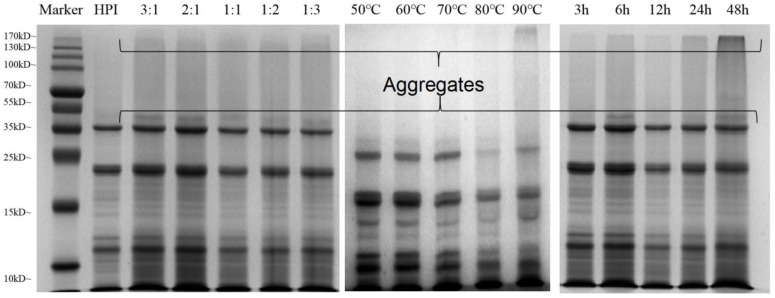
SDS-PAGE analysis of native hemp protein and glycosylated HPI–pullulan prepared under different conditions.

**Figure 3 foods-12-01416-f003:**
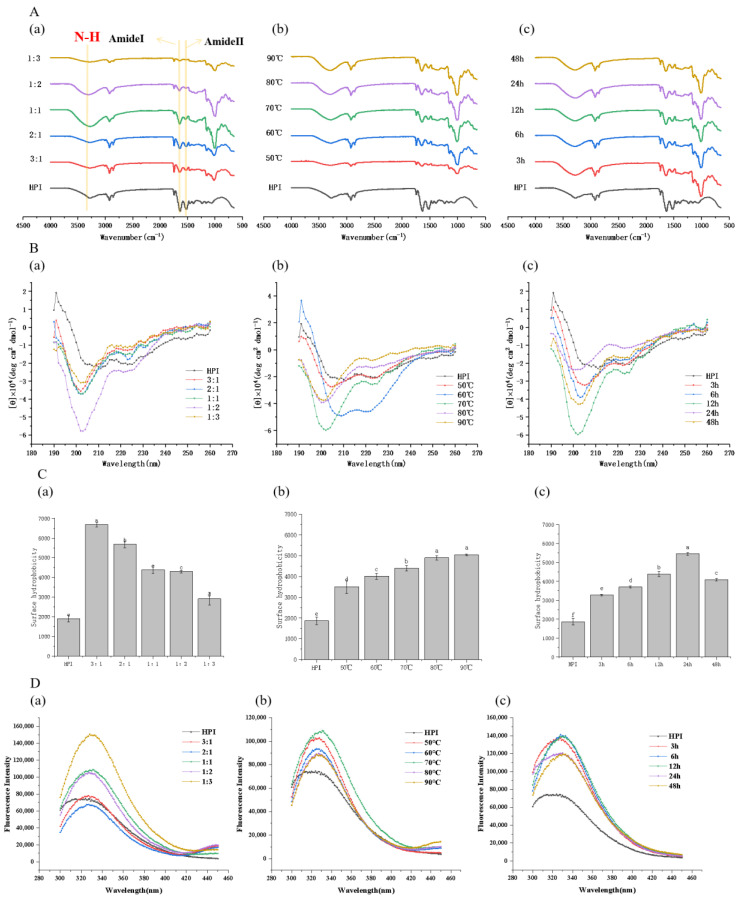
Effect of pullulan, ratios to hemp protein isolate (HPI), incubation temperature, and incubation time on spectroscopic analysis of glycosylated HPI: (**A**) FT-IR spectrum of HPI and HPI–pullulan conjugate; (**B**) secondary structure of HPI and HPI–pullulan conjugate; (**C**) the surface hydrophobicity (H0) of HPI and HPI–pullulan conjugate; (**D**) intrinsic fluorescence emission spectra of HPI and HPI–pullulan conjugate. (**a**): material feeding ratio (pH 7.0 and heated at 70 °C for 12 h); (**b**): reaction tempetature (pH 7.0 and heated 12 h with a HPI to pullulan ratio of 1:1); (**c**) reaction time (pH 7.0 and heated at 70 °C with a HPI to pullulan ratio of 1:1). Different superscript letters represent a significant difference (*p* < 0.05).

**Figure 4 foods-12-01416-f004:**
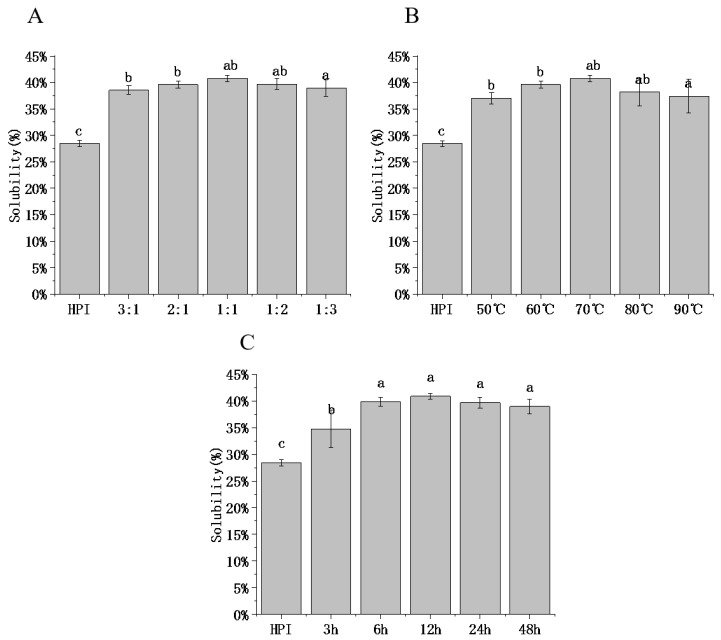
Effect of hemp protein isolate (HPI) ratios to pullulan (**A**), incubation temperature (**B**), and incubation time (**C**) on the solubility of glycosylated HPI. Different superscript letters represent a significant difference (*p* < 0.05).

**Figure 5 foods-12-01416-f005:**
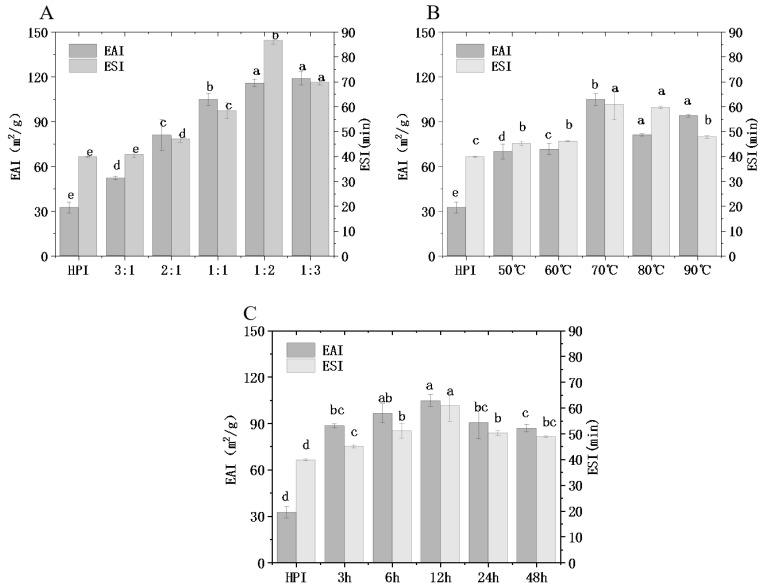
Effect of hemp protein isolate (HPI) ratios to pullulan (**A**), incubation temperature (**B**), and incubation time (**C**) on the emulsification ability of glycosylated HPI. Different superscript letters represent a significant difference (*p* < 0.05). EAI: emulsifying activity index. ESI: emulsion stability index.

**Figure 6 foods-12-01416-f006:**
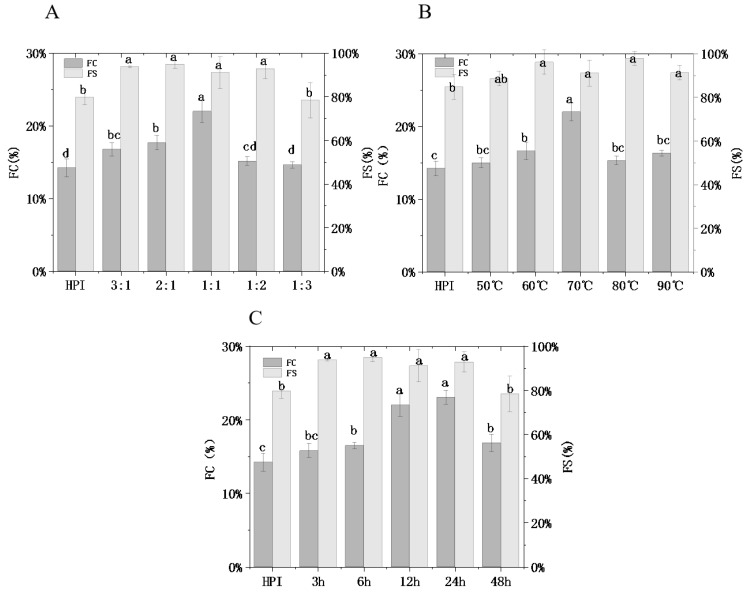
Effect of hemp protein isolate (HPI) ratios to pullulan (**A**), incubation temperature (**B**), and incubation time (**C**) on the foaming properties of glycosylated HPI. Different superscript letters represent a significant difference (*p* < 0.05). FC: foaming capacity. FS: foaming stability.

**Figure 7 foods-12-01416-f007:**
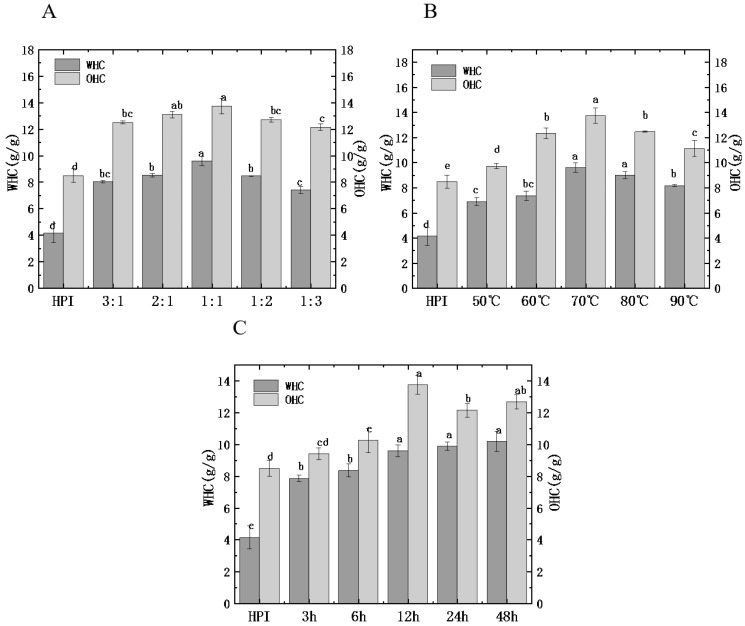
Effect of hemp protein isolate (HPI) ratios to pullulan (**A**), incubation temperature (**B**), and incubation time (**C**) on the water and oil absorption capacity (WAC/OAC) of glycosylated HPI. Different superscript letters represent a significant difference (*p* < 0.05). WHC: water holding capacity. OHC: oil holding capacity.

**Figure 8 foods-12-01416-f008:**
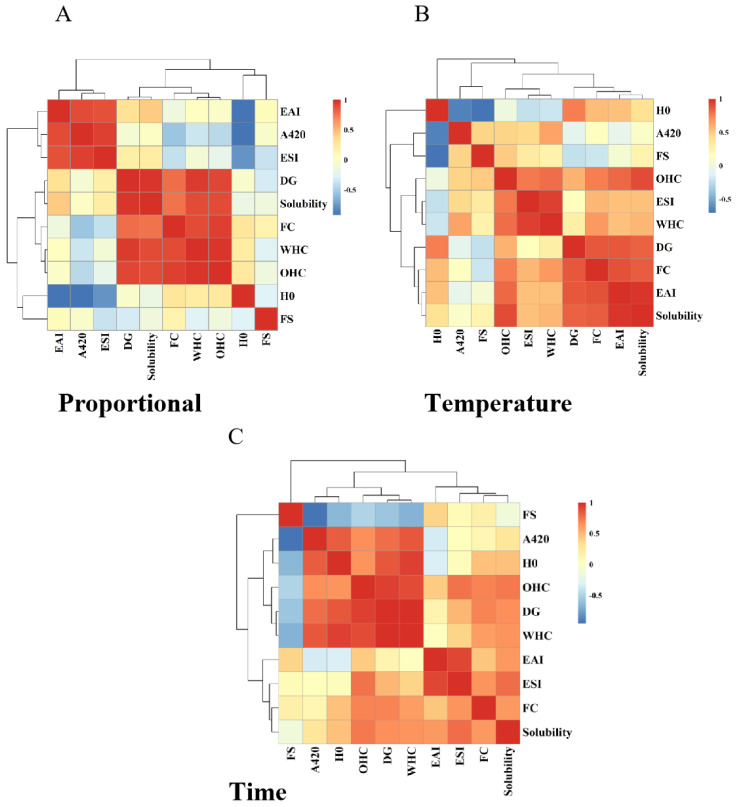
Correlation analysis between grafting (DG) and the functional properties of HPI–pullulan conjugate under different reaction conditions. (**A**): Analysis at different substrate ratios; (**B**): Analysis at different reaction tempetature; (**C**) Analysis at different reaction time.

## Data Availability

Data are contained within the article.

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
