# Peer review of "Structural and Functional Characteristics of Hemp Protein Isolate–Pullulan Polysaccharide Glycosylation Conjugate in an Aqueous Model System"

_foods, 2023, doi:10.3390/foods12071416_

Round 1

Reviewer 1 Report

This work presents Structural and functional characteristics of hemp protein isolate-pullulan polysaccharide glycosylation conjugate in an aqueous model system. This manuscript (MS) title is interesting.

Paper format: This paper is in required FOOD format.

Title: Fine

Abstract:  The abstract need revision in terms of including significant values among different data. Avoid theoretical approach in abstract, just show what results are saying. Keep to the point and address novelty of the work. The abstract should always be concise and informative. The arguments of why your study is important not making any sense. Extensive revision is required in abstract, as the present sentences sounds noisy during reading. Overall the abstract is not informative enough and need to show the actual picture of the work. Authors are supposed to please indicate the numerical values with significant difference in abstract. Keep abstract minimum with covering all of your data. For example see the sentence ‘’The current study revealed that glycosylation formed higher molecular weight conjugate in comparison with 20 non-glycosylated HPI as shown in DS-PAGE’’. This is wrong and many more in abstract. Please rewrite the abstract.

Keywords: Please change it to aims of the study, be specific

Introduction: Some shortcomings are below

1.     Lines 31 and 32, please justify this sentence with recent articles, how people demand for protein food? In what type of food?

2.     Please indicate what type of properties need to improve from hemp protein isolate? And what about hemp protein concentrates?

3.     Highlight some biological advantages of protein and saccharide complexes.

4.     Add application of protein glycation with more recent published papers.

5.     In my opinion the introduction section is too short.

Methods and materials:

1.     Section 2.1 split it in two parts, the isolation of protein must be clearly present with justified protocols.

2.     No color changed after glycation? Report it

3.     Free dried the sample or use liquid for further analysis? Explain here, as authors claimed in abstract and introduction the liquid phase

4.     What was the purity of protein?

5.     It’s better to examine the functional properties i.e. emulsion characterization with rheology and droplet sizes.

6.     Why TGA or DSC has not been determined for stabilities of complexes?

Results and discussion: Fair and as per figures, but I am surprise to see EAI and ESI values reported was much higher as compared to already available articles. I suggest to double check the results or recalculate it. So, this true then must compare with standard protein i.e. soyabean protein isolate for better understanding of results.

Conclusion: In conclusion add more of application of the products. Indicate what type of products are possible from these products.

Author Response

Dear Reviewer 1,

Manuscript ID: foods-2256671
Title: Structural and functional characteristics of hemp protein isolate-pullulan polysaccharide glycosylation conjugate in an aqueous model system

On behalf of my co-authors, we thank you very much for giving us an opportunity to revise our manuscript, we appreciate very much for your constructive comments and suggestions on our manuscript. We have tried our best to revise the manuscript according to your kind advice and detailed suggestions. Enclosed please kindly find the responses to the kind advice and detailed suggestions, and all revisions marked with red in the revised manuscript. We sincerely hope this manuscript will be finally accepted by the Foods. Thank you very much for all your help and looking forward to hearing from you soon.

Best regards
Sincerely yours
Yong-Jiang Xu, PhD, Professor
E-mail: yjxutju@gmail.com

Reviewer 2 Report

Totally speaking, this research article regarding the improvement of hemp protein techno-functional characteristics via glycosylation modification, is very well designed. Variations in the protein-to-polysacharide ration, temperature, and incubation time were used to optimize the conjugation process. The results demonstrate that glycosylation may loosen the secondary and tertiary protein structures, enhance the hydrophobicity of the protein surface and improved techno-functional charactoeristics (emulsifying activity, solubility, foaming ability, water and oil holding capacity). The authors performed the correlation analysis and emphazised that protein functional characteristics were positively correlated with the degree of glycosylation. The manuscript was suitable for Foods journal, and proposed special issue. However, there are a few minor points that require clarification.

(1) Affiliations of authors: In accordance with the guidelines for the preparation of the manuscript, the email addresses of all the authors, together with their initials in parenthesis, should be entered. Complete the required fields.

(2) Keywords: Please rewrite the terms ''physicochemical properties'' into ''Techno-functional properties'', and ''hempseed'' into ''hemp protein isolate''. Also, add the novel phrase ''protein-polysacharides complex''. 

(3) Introduction: Please complete the introduction with specific statistical data on the worldwide utilization and production hemp proteins and its products. Next, isolate the procedures for obtaining hemp protein fractions with improved functional chracteristics.

There is a lack of a review of the literature on the subject of the current ways of improving the properties of hemp protein in combination with polysaccharides. So, which polysaccharides have been investigated so far, and what kind of results have been found? Complete the introduction with the necessary data.

Please provide a few more research goals at the conclusion of the introductory section. These goals will be divided into phases based on the results of the carried out experimental studies. Hence, the primary goal of this study will be made clear.

(4) Materials and Methods section: The names of the devices (centrifuges, reactors, freeze-dryer, etc.) that were used in the experimental work and their manufacturers must be mentioned.

Reference the Kjeldahl method and used covnersion factor; specify the utilized micro or macro Kjeldal device.

Was the glycosylation process performed in an appropriate water bath, incubator, or dryer? Fill in the required.

Line 111: What fluorescent dye was used to measure surface hydrophobicity? What concentration of samples was prepared for analysis?

It is necessary to at the very least provide a brief description of the techniques employed to explain the performance of analyses for functional and structural features. Referencing the pertinent literature alone is insufficient. Thus, first explain how to construct a sample of a protein-polysaccharide conjugate for the related analysis, and then quickly summarize the performance.

(5) Results and Discussion: According to the guidelines given to the writers, the findings should be shown more graphically and each diagram's resolution should be increased.

(6) It is advised that the authors recheck the main text during the revision to make this manuscript more readable. Furthermore, pay attention to the list of references, doi numbers are missing and in some places complete pagination.

Author Response

Dear Reviewer 2,

Manuscript ID: foods-2256671

Title: Structural and functional characteristics of hemp protein isolate-pullulan polysaccharide glycosylation conjugate in an aqueous model system

On behalf of my co-authors, we thank you very much for giving us an opportunity to revise our manuscript, we appreciate very much for your constructive comments and suggestions on our manuscript. We have tried our best to revise the manuscript according to your kind advice and detailed suggestions. Enclosed please kindly find the responses to the kind advice and detailed suggestions, and all revisions marked with red in the revised manuscript. We sincerely hope this manuscript will be finally accepted by the Foods. Thank you very much for all your help and looking forward to hearing from you soon.

Best regards

Sincerely yours

Yong-Jiang Xu, PhD, Professor

E-mail: yjxutju@gmail.com

Round 2

Reviewer 1 Report

The matter which was concern to me was addressed properly, I am agree for acceptance of the manuscript.